# Monoallelic *CRMP1* gene variants cause neurodevelopmental disorder

Ethiraj Ravindran[1,2,3†], Nobuto Arashiki[4†], Lena-Luise Becker[1,2,3‡],
Kohtaro Takizawa[4‡], Jonathan Lévy[5,6], Thomas Rambaud[6],
Konstantin L Makridis[1,2,3], Yoshio Goshima[7], Na Li[8], Maaike Vreeburg[9],
Bénédicte Demeer[10,11], Achim Dickmanns[12], Alexander PA Stegmann[9], Hao Hu[8*],
Fumio Nakamura[4*], Angela M Kaindl[1,2,3*]

[1]Department of Pediatric Neurology, Charité - Universitätsmedizin Berlin, Berlin, Germany; [2]Center for Chronically Sick Children, Charité–Universitätsmedizin Berlin, Berlin, Germany; [3]Institute for Cell Biology and Neurobiology, Charité–Universitätsmedizin Berlin, Berlin, Germany; [4]Department of Biochemistry, Tokyo Women's Medical University, Tokyo, Japan; [5]Department of Genetics, Robert Debré University Hospital, Paris, France; [6]Laboratoire de biologie médicale multisites Seqoia, Paris, France; [7]Department of Molecular Pharmacology and Neurobiology, Graduate School of Medicine, Yokohama City University, Yokohama, Japan; [8]Laboratory of Medical Systems Biology, Guangzhou Women and Children's Medical Center, Guangzhou Medical University, Guangzhou, China; [9]Clinical Genetics, Maastricht University Medical Centre, Maastricht, Netherlands; [10]Center for Human Genetics, CLAD Nord de France, CHU Amiens-Picardie, Amiens, France; [11]CHIMERE EA 7516, University Picardie Jules Verne, Amiens, France; [12]Department of Molecular Structural Biology, Institute for Microbiology and Genetics, Georg-August-University Göttingen, Göttingen, Germany

**\*For correspondence:**
huh@cougarlab.org (HH);
nakamura.fumio@twmu.ac.jp
(FN);
angela.kaindl@charite.de (AMK)

†These authors contributed
equally to this work
‡These authors also contributed
equally to this work

**Competing interest:** The authors declare that no competing interests exist.

**Abstract** Collapsin response mediator proteins (CRMPs) are key for brain development and function. Here, we link CRMP1 to a neurodevelopmental disorder. We report heterozygous de novo variants in the *CRMP1* gene in three unrelated individuals with muscular hypotonia, intellectual disability, and/or autism spectrum disorder. Based on in silico analysis these variants are predicted to affect the CRMP1 structure. We further analyzed the effect of the variants on the protein structure/levels and cellular processes. We showed that the human *CRMP1* variants impact the oligomerization of CRMP1 proteins. Moreover, overexpression of the *CRMP1* variants affect neurite outgrowth of murine cortical neurons. While altered CRMP1 levels have been reported in psychiatric diseases, genetic variants in *CRMP1* gene have never been linked to human disease. We report for the first-time variants in the *CRMP1* gene and emphasize its key role in brain development and function by linking directly to a human neurodevelopmental disease.

## Editor's evaluation

The authors report the first human disease-related variants in CRMP1 leading to a neurodevelopmental syndrome. After a successful first review, the authors have addressed the reviewer comments fully and have carefully assessed and fulfilled the criteria for essential revisions with the inclusion of new data. The results will be of importance to cell biologists and clinicians.

## Introduction

Neurodevelopment is a fine-tuned process orchestrated by distinct expression and function of several genes and any disturbances in this timely controlled process culminate in neurodevelopmental disorder (*Rice and Barone, 2000*; *Gilbert et al., 2005*). Collapsin response mediator proteins (CRMPs) are cytosolic phosphoproteins that are highly and differentially expressed in the nervous system (*Wang and Strittmatter, 1996*; *Bretin et al., 2005*). The five CRMP subtypes (CRMP1–5) form homo- or hetero-tetramers in various combinations and thereby enable distinct functions key for neurodevelopment (*Wang and Strittmatter, 1997*). Targeted neurodevelopmental processes include cell migration, axonal outgrowth, dendritic branching, apoptosis mediated through extracellular signaling molecules (Sema3A, reelin, neurotrophins) (*Yamashita et al., 2006*; *Su et al., 2007*; *Charrier et al., 2006*; *Yamashita et al., 2007*; *Makihara et al., 2016*). CRMP function is regulated in a spatiotemporal manner through protein phosphorylation mediated by various kinases such as Cdk5, Rho/ROCK, and GSK3 (*Yoshimura et al., 2005*; *Uchida et al., 2005*).

Given their key function in developmental processes, disturbances in CRMP function can result in neurodevelopmental diseases. In this line, monoallelic *CRMP5* variants can cause Ritscher–Schinzel syndrome 4 (MIM#619435), a neurodevelopmental disease with craniofacial features, cerebral and cardiovascular malformations, and cognitive dysfunction (*Jeanne et al., 2021*). *CRMP4* variants have been associated with amyotrophic lateral sclerosis in the French population (*Blasco et al., 2013*).

Here, we link for the first-time *CRMP1* variants in three unrelated pedigrees to neurodevelopmental disorder in humans with muscular hypotonia, autism spectrum disorder (ASD), and/or intellectual disability. In humans, maternal CRMP1 autoantibodies have been associated with autism in their children, and increased *CRMP1* mRNA levels were identified in individuals with schizophrenia, attention-deficit hyperactivity disorder, and ASD (*Braunschweig et al., 2013*; *Bader et al., 2012*). Knockout of *Crmp1* in mice results in schizophrenia-associated behavior, impaired learning and memory, and prepulse inhibition (*Yamashita et al., 2013*). At the cellular level, abnormal neurite outgrowth, dendritic development and orientation, and spine maturation of cortical and/or hippocampal neurons have been shown (*Su et al., 2007*; *Yamashita et al., 2007*; *Makihara et al., 2016*). Loss of *Crmp1* affects long-term potentiation maintenance (*Su et al., 2007*). In addition, its loss during cerebellar development leads to reduced granule cell proliferation, apoptosis, and migration in the cerebellum of *Crmp1*$^{-/-}$ mice (*Charrier et al., 2006*). Although evidence on altered levels of CRMP1 in neuropsychiatric diseases exists, the variants in the *CRMP1* gene have not been linked to a human disease.

## Results

### Phenotype and genotype of index patients

Proband 1 (P1) was born as the second child of non-consanguineous healthy parents of Caucasian descent after an uneventful pregnancy (*Figure 1A*, *Table 1*). At delivery a singular umbilical artery was noted. The global development was delayed from infancy on: with sitting at 18 months, standing with support at 14 months, walking at 26 months, first words at 24 months. She had slurred speech. Standardized cognitive tests performed at 5 and 10 years revealed a moderate intellectual disability with an intelligence quotient (IQ) of 55 at last assessment using the Kaufman Assessment Battery for Children (K-ABC). Behavioral problems included a lack of distance to men and a sexualized behavior. At last assessment at 16 years-of-age, the girl had generalized muscular hypotonia with normal reflexes, but fine motor problems with a broad-based gait but no ataxia. When climbing stairs, she showed clear instability. The results of cranial magnetic resonance imaging (MRI) at 1.5 and 4 years as well as that of further work-up (including metabolic tests, electroencephalogram [EEG], ophthalmological assessment, electrocardiogram, echocardiogram, and abdominal sonography) were normal. Since preliminary genetic tests including chromosome analysis and array-CGH were normal in the index patient (P1), we performed whole-exome sequencing (WES) to identify the underlying genetic cause. WES followed by bioinformatic analysis and confirmation with Sanger sequencing revealed the heterozygous de novo variant in the *CRMP1* gene c.1766C>T (NM_001014809.2; Chr4 (GRCh37):g.5830253G>A) in the affected child P1 (*Figure 1B*; *Figure 1—figure supplement 1*). At the protein level, this variant leads to an amino acid change at position 589 from proline to leucine: P589L in long isoform, CRMP1A (NP_001014809.1), and P475L in short form, CRMP1B (NP_001304.1)

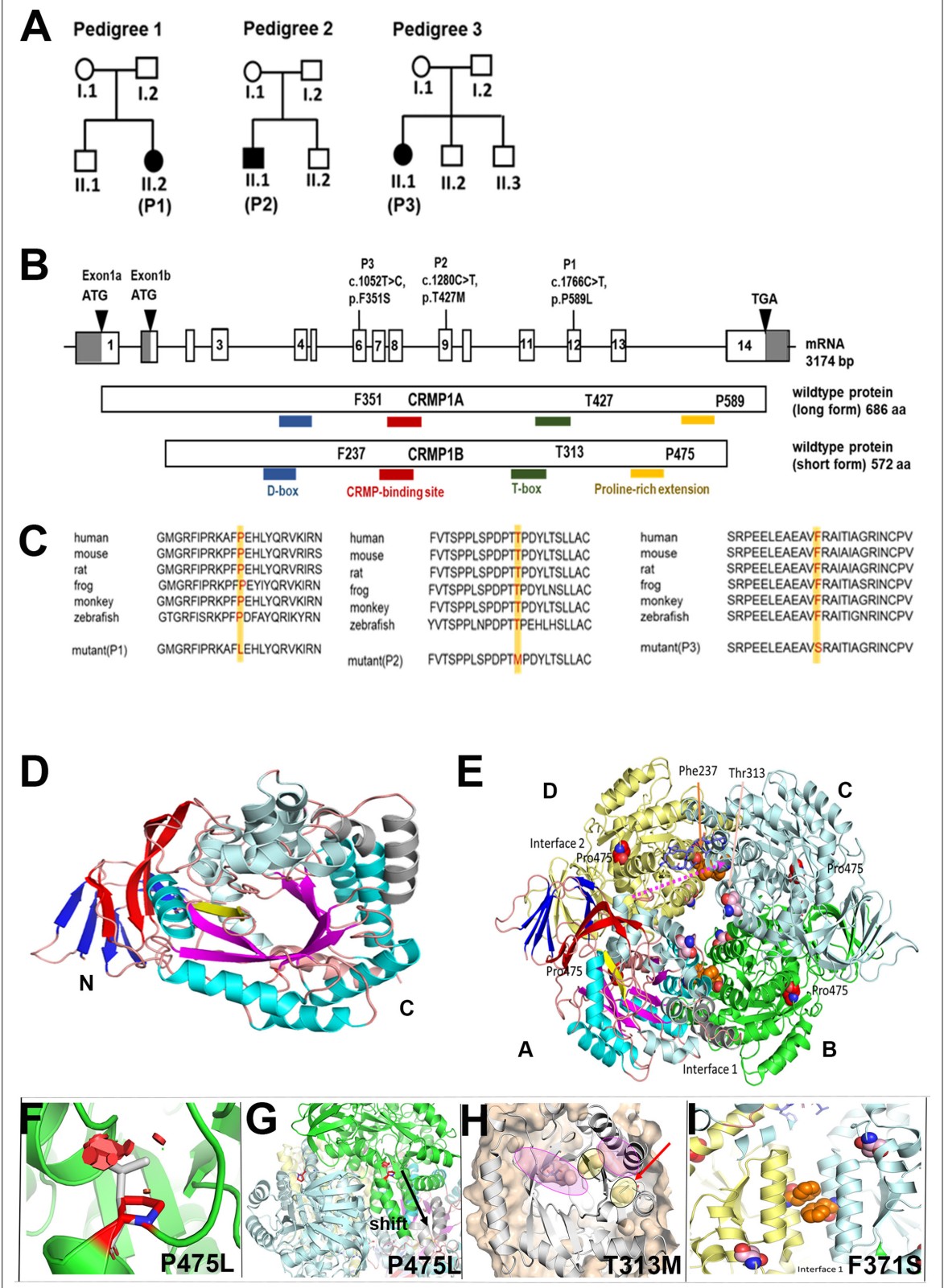

**Figure 1.** Genotype of patients with variants in *CRMP1*. (**A**) Pedigree of index families. (**B**) Pictogram representing the CRMP1 cDNA with identified variant of proband 1 (P1) in exon 12 (c.1766C>T, NM_001014809.2) which leads on protein level to an amino acid change of proline to leucine in CRMP1 (CRMP1A-long form (p.P589L, NP_001014809.1) and CRMP1B-short form (p.P475L, NP_001304.1)); the variant in proband 2 (P2) in exon 9 c.280C>T (NM_001014809.2) leads to an exchange of threonine to methionine (CRMP1A (p.T427M, NP_001014809.1) and CRMP1B (p.T313M, NP_001304.1)); the

*Figure 1 continued on next page*

*Figure 1 continued*

variant in proband 3 (P3) in exon 6 c.052T>C (NM_001014809.2) leads to an exchange of phenylalanine to serine (CRMP1A (p.(F351S), NP_001014809.1) and CRMP1B (p.(F237S), NP_001304.1)). (**C**) Multispecies sequence alignment localizes the variants in the highly conserved area of CRMP1. (**D**) The short form CRMP1 monomer is composed of three structural parts, an N-terminally located seven β-strands forming two β-sheets (depicted in blue), followed by a linker β-strand (yellow) connecting to the central α/β-barrel (cyan/magenta) formed by seven repeats. Inserted after repeat 4 are 2 additional α-helices (gray). (**E**) CRMP1 assembles into tetramers. The relevant sites of T313M, P475L, and F237S are indicated as sphere model, with the T313 and F237 located in the central channel in the vicinity of the interaction sites and P475 is located at the beginning of the C-terminal helix and oriented toward the adjacent molecules. (**F**) The variant P475L reveals serious clashes with neighboring residues (red hexagonals) which may be accounted for by a shift of the helix as shown in (**G**). (**H**) Detailed representation of the structural vicinity of the T313M (yellow) to the ligand-binding cavity (magenta). (**I**) Magnified view of interface 1 tilted 90° backwards with respect to panel E highlighting the arrangement of the two phenylalanines at position 237 from the neighboring units. The exchange of phenylalanine with hydrophobic residues to serine with hydrophilic side chain interferes with the stability of the interaction in interface 1.

The online version of this article includes the following figure supplement(s) for figure 1:

**Figure supplement 1.** Sanger sequencing of *CRMP1* in pedigrees 1 and 3.

(*Figure 1B*). The identified variant was not found in 1000Genomes, dbSNP, or gnomAD. The CADD phred score (https://cadd.gs.washington.edu/) and REVEL score was 24.8. and 0.695, respectively. The variant localizes to a highly conserved position, as demonstrated by PhyloP (5.327), PhastCons (1) score, and multispecies sequence alignment (*Figure 1C*).

Proband 2 (P2) was born as the second child of non-consanguineous parents of Caucasian descent after an uneventful pregnancy and delivery (*Figure 1A*, *Table 1*). Routine echographia during pregnancy did not show of any concern. The boy was macrosomic at birth but had no congenital microcephaly. His motor development was delayed (unsupported walking at 24 months-of-age) due to congenital mild muscular hypotonia with normal deep tendon reflexes, but no coordination problems. He had bilateral pes planus. A speech delay and language impairment with first words spoken at 36 months were diagnosed. He was also diagnosed with an ASD and normal cognitive abilities (IQ 95). At last assessment at 10 years-of-age, an obesity associated with hyperphagia was of raising concern, and he had secondary enuresis nocturna. The body mass index (BMI) at last evaluation was 26.1. The results of a cranial MRI and EEG were normal. Routine metabolic screening showed no abnormal results. Similarly, through WES followed by bioinformatic analysis, we identified a heterozygous de novo variant c.1280C>T in the *CRMP1* gene (NM_001014809.2; Chr4 (GRCh37):g.5841279G>A) in proband (P2) (*Figure 1B*). This variant leads to an exchange of threonine to methionine at position 427 in the long form, CRMP1A (p.T427M, NP_001014809.1) and at position 313 in the short form, CRMP1B (p.T313M, NP_001304.1) (*Figure 1B*). The variant affects a highly conserved region of the protein (*Figure 1C*). The CADD phred score and REVEL score were 27.7 and 0.819, respectively. PhyloP (5.093), PhastCons (1) score, and multispecies sequence alignment show that the variant is in a highly conserved region (*Figure 1C*).

Proband 3 (P3) is the first child of three of a non-consanguineous family of European descent (*Figure 1A*, *Table 1*). She was born at gestation week 40 after an uneventful pregnancy, with normal birth parameters (weight: 3.610 kg, height: 50 cm, head circumference: 32.5 cm, Apgar score: 10/10). She had developmental delay with not being able to sit alone at 11 months and walked without support at 24 months. Her speech development was delayed with few words spoken at 30 months. She began to gain weight from 18 months, and overgrowth was noticed since the age of 2 years. At the age of 6 years, cell blood count, ionogram, lipid profile, endocrinological screening, and leptin blood level were normal with low-density lipoprotein (LDL) cholesterol: 3.16 mmol/l (2.60–4.00), total cholesterol/high-density lipoprotein (HDL) cholesterol ratio: 3.9 mmol/l (<4.4), serum triglycerides: 0.47 mmol/l (0.42–1.40), HbA1C: 5.4% (4.0–6.0), thyroid stimulating hormone (TSH): 3.27 µUI/ml (0.15–3.70), FT4: 10.25 ng/l (6.10–11.20), Cortisolemia: 92 µg/l (7.87–14.45), and Leptinemia: 131 ng/ml (19–251). There is familial history of obesity on both parental sides, and the father is macrocephalic (head circumference: 60.5 cm). She developed severe behavioral issues with temper tantrums, stubbornness, hyperphagia, obsessive–compulsive characteristics and ASD. She was attending special educational school at the age of 8. At the last assessment of 13 years-of-age, she had moderate intellectual disability and persistent severe behavioral disorders. Distinctive facial features were low forehead hair insertion, anteverted and large earlobes, broad nasal tip, short philtrum, and full lower lips. Genu valgum and hyperlordosis as well as abdominal and dorsal stretch marks were noted. Enuresis

**Table 1.** Phenotype of patients with *CRMP1* variants.

| Characteristics and symptoms | Proband 1 (pedigree 1) | Proband 2 (pedigree I1) | Proband 3 (pedigree I1I) |
|---|---|---|---|
| *CRMP1* variant (NM_001014809.2); (NP_001014809.1) | c.1766C>T; p.P589L | c.1280C>T; p.T427M | c.1052T>C; p.(F351S) |
| Parents | Non-consanguineous | Non-consanguineous | Non-consanguineous |
| Gender | Female | Male | Female |
| Age at last assessment (years) | 15 | 10 | 13 |
| Anthropometric data | Normal | Normal | Overgrowth |
| Weight (kg) | 57.1 (0.15 SD) | | 133 (14.2 SD) |
| Height (cm) | 168 (0.43 SD) | N/A | 166.5 (2.94 SD) |
| OFC (cm) | 53.5 (−1.02 SD) | | 62 (5 SD) |
| Pregnancy, birth, postnatal adaption | Normal | Normal | Normal |
| singular umbilical artery | + | − | − |
| macrosomia | − | + | − |
| fetal fingerpads | − | + | − |
| Microcephaly (OFC <−2 SD) | − | − | − |
| Macrocephaly (OFC >−2 SD) | − | − | + (5 SD) |
| Facial dysmorphism | − | − | + |
| Delayed motor development | + | + | + |
| walking unsupported | 28 months | 24 months | 24 months |
| Global muscular hypotonia | Mild | Mild | Mild |
| Deep tendon reflexes | Normal | Normal | Normal |
| Intellectual disability | Moderate (IQ 55) | No (IQ 95) | Moderate |
| Autism spectrum disorder | − | + | − |
| Behavioral problems | + | + | + |
| lack of distance, sexualized behavior | + | − | − |
| hyperphagia | − | + | + |
| Delayed speech and language development (first words spoken) | + (24 months) | + (36 months) | + (30 months) |
| Fine motor problems | + | − | + |
| Other | | | |
| secondary enuresis | | | |
| Nocturna | − | + | + |
| pes planus | − | + | − |
| obesity | − | + | + |
| EEG results | Normal | Normal | Normal |
| Cranial MRI abnormalities | − | − | − |

+, yes; −, no; IQ, intellectual quotient; N/A, not available; OFC, occipitofrontal circumference; SD, standard deviation.

was noted. Cerebral MRI and abdomino-renal ultrasound were normal. Metabolic disorder screening and storage disease screening were negative, serum insulinemia was elevated: 32 mUI/ml (<25) with normal glycemia (0.9 g/l). Array analysis revealed two maternally inherited deletions: a 668 kb deletion at 3q26.31 and a 371 kb at 5q23.1, confirmed by genome sequencing and considered as variant of unknown significance. Further analysis through trio-based whole-genome sequencing identified a de novo variant in the *CRMP1* gene c.1052T>C (NM_001014809.2; Chr4 (GRCh37):g.5841409A>G) (*Figure 1B*) and the mutation was confirmed by Sanger sequencing (*Figure 1—figure supplement 1*). The identified variant leads to an exchange of phenylalanine to serine at position 351 of long form of CRMP1 (CRMP1A (p.(F351S), NP_001014809.1)) and at position 237 in short form (CRMP1B (p.(F237S), NP_001304.1)) and it is located in the highly conserved region of the protein (*Figure 1C*). This variant is not found in gnomAD database. In silico pathogenicity prediction tools predict the identified variants to be deleterious (CADD phred: 23.60; REVEL 0.577), and the variant is located in the highly conserved region (PhyloP (2.826), PhastCons (1)). No additional pathogenic variants have been identified by trio genome sequencing, including all known genes involved in neurodevelopmental disorder.

## Effect of identified *CRMP1* variants on protein structure

Since CRMP1 is known to oligomerize to form homotetramers and heterotetramers along with other CRMPs to regulate cellular functions (*Wang and Strittmatter, 1997*), we determined the effect of the identified human variants on its protein structure using known structures and protein structure prediction tools. The structure of CRMP1 short form, CRMP1B, has been determined at 3.05 A resolution (*Liu et al., 2015*) lacking the N-terminal 14 residues and residues 491–572 in their C-terminal region. A CRMP1 monomer consists of three structural parts, N-terminal β-strands followed by a linker β-strand connected to the central α/β-barrel (*Figure 1D*). Several residues contribute to the oligomerization interface of CRMP1 and the quaternary structure of CRMP1 tetramer is shown in *Figure 1E*. The amino acids (P475, T313, and F237) are located in the highly conserved region of CRMP1. The P475 is the last residue before the start of the C-terminal helix and the exchange of proline to leucine (p.(P475L)) is predicted to lead to serious clashes with the neighboring residues (*Figure 1F*). Such spatial constraints may alter the dipole moment of the helix and lead to long-range allosteric effects (*Figure 1G*). The T313 lies within the region of the α/β-barrel and oriented toward the inside of the protein and located close to one of the two cavities (highlighted in yellow) (*Figure 1H*). These two cavities are located in between two pockets (highlighted in magenta). Conformational changes within these pockets upon function or interaction with other molecules could shuffle these cavities to form a channel required for function. The exchange of threonine to methionine (p.(T313M)) is predicted to prevent the conformational changes which might affect the interaction of CRMP1 with other molecules and/or oligomerization. The F237 is localized directly in the center of interface 1 so the exchange of a heavy hydrophobic sidechain by a hydrophilic short sidechain (p.(F237S)) most likely interferes with the stability of the dimer interaction (*Figure 1I*). Based on structural simulations, all three variants are predicted to affect the ternary structure of CRMP1 and impact on its oligomerization.

## P475L and T313M variants affect homo-oligomerization of CRMP1B

To analyze the effect of *CRMP1* variants on its protein levels and cellular function, two variants (CRMP1B-P475L (P1) or -T313M (P2)) were chosen for further functional analysis. The alternate splicing of exons 1a and 1b of *CRMP1* gene brings two splicing variants CRMP1A (686 aa, long form) and CRMP1B (572 aa short form) (*Figure 1B*). CRMP1B is the major isoform expressed in the nervous system and most of CRMP1 studies including X-ray structural analysis have been performed with CRMP1B and therefore we used CRMP1B isoform for the experiments.

We purified the recombinant human CRMP1B-wildtype, -P475L (P1), or -T313M (P2) proteins using the *E. coli* GST-tag expression system. CRMP1B-wildtype showed two major 64 and 60 kDa bands on SDS-denatured gel electrophoresis (*Figure 2A*, left lane). The 64 and 60 kDa correspond to full-length CRMP1B and to a truncated, C-terminal region cleaved form, respectively. The yield of purified T313M (TM) or P475L (PL) was less than that of wildtype in the same condition (*Figure 2A*, middle and right lanes). This finding may be due to lower expression and/or aggregation of the variant proteins in *E. coli*. As T313M and P475L mutated residues are positioned close to the dimer/tetramer interface of CRMP1B, these variants may affect homo-oligomerization of CRMP1. We therefore examined the

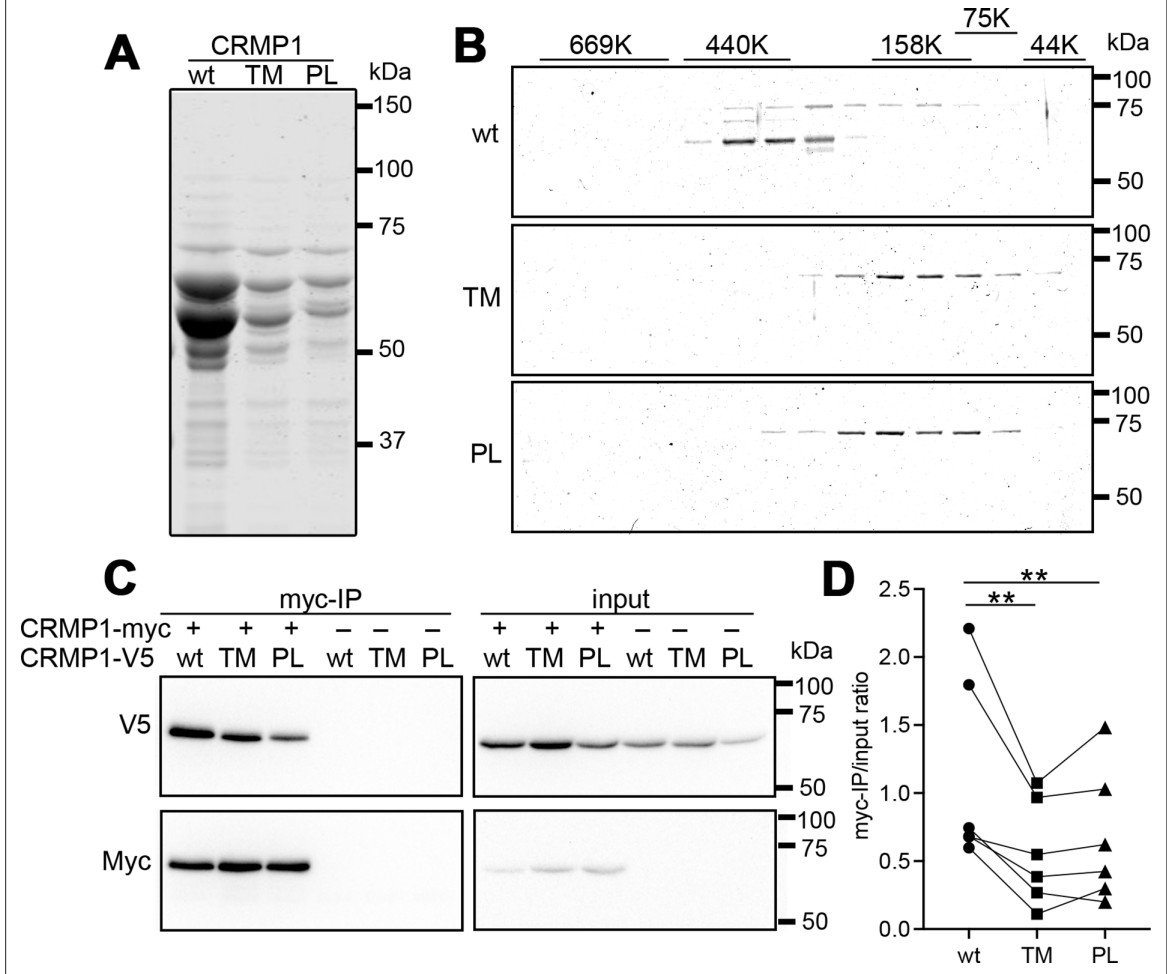

**Figure 2.** Attenuated oligomer formation of CRMP1B-P475L and CRMP1B-T313M variants. (**A**) Purified CRMP1B-wildtype, -T313M, and -P475L recombinant proteins on sodium dodecyl sulfate–polyacrylamide gel electrophoresis (SDS–PAGE) stained with CBB. GST-tagged CRMP1 were expressed in *E. coli* and purified through the binding to glutathione resin and the digestion with PreScission protease. Equal volume (2.5 µl) of the purified specimens, which were prepared under the same condition, were loaded on the gel. The yield of the variants was less than that of CRMP1-wildtype as shown in panel. Two major 64 and 60 kDa bands in the preparations are full-length and a truncated form, respectively. (**B**) Fractionation of CRMP1B specimens by size-exclusion chromatography. Purified CRMP1B specimens (150 µg) were passed through a Sephacryl S-300 column and the resultant flows/elution volumes were fractionated at every 1 ml. SDS–PAGE for 43–77th fractions were carried out after 20 times concentration. The distribution of molecular weights of standard proteins (44–660 kDa) at the same condition was displayed on the top of the gels. (**C**) Reduced homophylic interaction of CRMP1B variants. HEK293T cells coexpressing Myc-tagged CRMP1B-wildtype and either one of V5-tagged CRMP1B-wildtype, -T313M, or -P475L were analyzed by co-immunoprecipitation with anti-Myc-antibody. Immunoprecipitated specimens and input lysates were subjected to anti-V5 and anti-Myc immunoblot analyses. Co-immunoprecipitation of V5-tagged CRMP1 variants were reduced comparing to of wildtype CRMP1-V5. (**D**) Quantification of the V5-signal of Myc-immunoprecipitated specimens and of input lysate. The V5-signal ratios of CRMP1B-T313M and CRMP1B-P475L were significantly decreased compared to the ratio of CRMP1B-wildtype. The graph represents V5-signal ratio of each condition from six independent experiments (n=6). Data were analyzed by one-way repeated measures analysis of variance (ANOVA) followed by Tukey's multiple comparisons test. **p < 0.01. Abbreviations: CRMP1B-wildtype, wt; CRMP1B-T313M, TM; CRMP1B-P475L, PL.

The online version of this article includes the following source data for figure 2:

**Source data 1.** CRMP1 variants impact the homo-oligomerization.

**Source data 2.** Ratio of signal intensity of myc-IP band and input ratio of V5 blot.

oligomerization of recombinant CRMP1 preparation with gel-filtration chromatography (*Figure 2B*). Truncated CRMP1B-wildtype (lower band) was present in the high molecular weight region (200–500 kDa) and full-length CRMP1B (higher band) was present between 400 and 100 kDa. The high molecular weight signal probably represents the homo-oligomerization of CRMP1B-wildtype under the native condition. In contrast, CRMP1B-T313M and -P475L were detected at the lower molecular

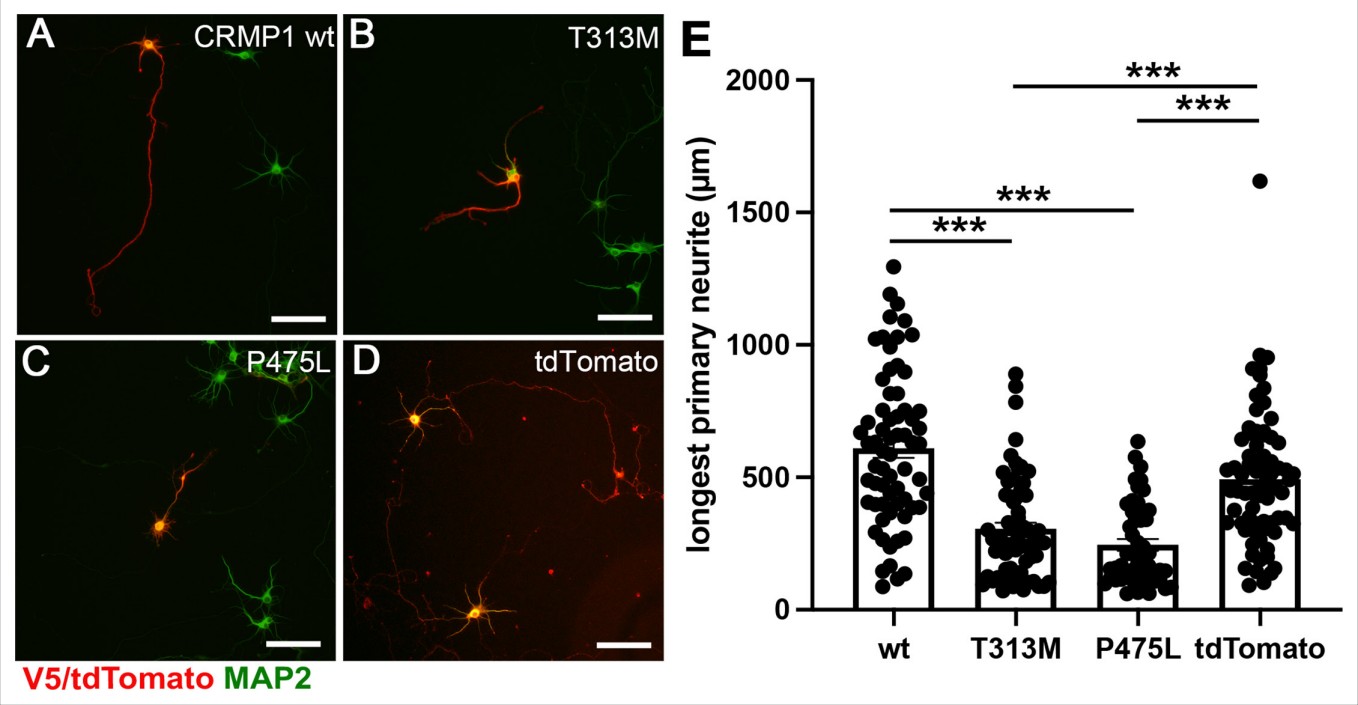

**Figure 3.** Attenuated neurite outgrowth by the ectopic expression of CRMP1B- P475L and CRMP1B-T313M variants. Representative images of the neurons expressing V5-CRMP1B-wildtype (**A**), -T313M (**B**), -P475L (**C**), or tdTomato (**D**). Transfected neurons were visualized by anti-V5 immunostaining or tdTomato expression (red) and anti-MAP2 immunostaining (green). The longest primary neurites of the neurons expressing V5-CRMP1B-T313M or -P475L were shorter than those of the neurons transfected with V5-CRMP1B-wildtype or tdTomato. Scale bars, 100 μm. (**E**) Longest primary neurite length. The length of the longest primary neurite from V5- or tdTomato-positive neurons was scored in each condition. The graph represents average ± standard error of the mean (SEM) with individual values from four independent experiments. The number (n) of examined neurons in each condition: CRMP1B-wildtype, 66; -T313M, 64; -P475L, 49; tdTomato, 73. Data were analyzed by one-way analysis of variance (ANOVA) followed by Tukey's post hoc test. ***p < 0.001.

The online version of this article includes the following source data for figure 3:

**Source data 1.** Quantification of longest neurite length.

weight region (40–200 kDa). The molecular size (120 kDa) of peak fractions of T313M and P475L may indicate the dimerization of these variant proteins.

We further examined the homophilic interaction of CRMP1B-wildtype and the variants. HEK293T cells were co-transfected with Myc-tagged CRMP1B-wildtype together with either one of V5-tagged CRMP1B-wildtype, CRMP1B-T313M, or CRMP1B-P475L. The cells were subjected to anti-Myc-immunoprecipitation. As shown in *Figure 2C*, while V5-tagged CRMP1B-wildtype was strongly co-immunoprecipitated with Myc-tagged CRMP1B-wildtype, co-immunoprecipitation of V5-tagged *CRMP1B*-T313M or -P475L was reduced. Immunoprecipitated and input V5-signal ratios from CRMP1B-T313M and CRMP1B-P475L were significantly decreased compared to CRMP1B-wildtype (*Figure 2D*). These results suggest that the amino acid exchange in the CRMP1 variants may interfere with the homophilic interaction of CRMP1 and homo-oligomerization.

## P475L and T313M variants attenuate neurite outgrowth of cortical neurons

We next asked whether the ectopic expression of *CRMP1B* variants in primary cultured murine cortical neurons affect the neuronal development. Dissociated E15 mouse cortical neurons were electroporated with the expression vector harboring either V5-*CRMP1B*-wildtype (wt), -T313M (TM), -P475L (PL), or tdTomato. The cells were seeded onto PLL-coated culture dishes and grown for 6–7 days. After fixation, the cells were immunostained with anti-V5 and anti-MAP2 antibodies. We found that the longest primary neurites of V5-*CRMP1B*-T313M or -P475L transfected neurons were shorter than those of V5-*CRMP1B*-wildtype (wt) or tdTomato transfected cells (*Figure 3A–D*). We then measured

the length of the longest primary neurite from V5- or tdTomato-positive neurons in each condition. The longest primary neurites expressing *CRMP1B*-T313M or -P475L were 40–50% shorter than those expressing CRMP1B-wildtype (wt) (*Figure 3E*). As the longest primary neurites of the cultured cortical neurons are thought to be axons, these CRMP1B variants may interfere with the oligomerization of CRMP1 in turn to attenuate the outgrowth of the cortical axons.

## Discussion

In this study, we report the human phenotype associated with heterozygous *CRMP1* variants in three affected children of unrelated non-consanguineous pedigrees. All the patients have a neurodevelopmental disorder with motor delay and muscular hypotonia. While patient P1 has moderate intellectual disability and behavioral abnormalities, P2 was diagnosed with an ASD but a normal cognitive profile, and P3 presented with moderate intellectual disability and ASD.

We showed that those human *CRMP1* variants impact the oligomerization of CRMP1 proteins. CRMP1 exists as homo- or hetero-tetramers to interact with various signaling molecules and to link microtubules to subcellular structures and regulate cytoskeletal dynamics (*Wang and Strittmatter, 1997*; *Nakamura et al., 2014*). In line with this, CRMP1 colocalizes to the mitotic spindles and centrosomes, indicating of playing a key role in regulating mitosis and cell cycle progression (*Shih et al., 2003*). Defective cell cycle, abnormal mitotic spindle and centrosomes are the common key pathomechanisms underlying several neurodevelopmental disorders (*Zaqout and Kaindl, 2021*).

Dysregulation of other CRMPs that have been linked to human disease has been shown to affect specific cellular processes associated with the clinical phenotype. For example, missense variants of CRMP5 impede the ternary complex formation with microtubule-associated protein 2 (MAP2) and beta-III-tubulin, thereby causing a defective inhibitory regulation on neurite outgrowth and dendritic development and thereby contribute to the brain malformation phenotype (*Jeanne et al., 2021*). MAP2 has been already reported to play a key role in neurite outgrowth and synaptic plasticity through its role in microtubule stabilization and cytoskeletal association (*Kaech et al., 2001*; *Sánchez et al., 2000*). Also, variant CRMP4 was shown to affect axonal growth and survival of motor neurons and thereby contribute to the amyotrophic lateral sclerosis phenotype (*Blasco et al., 2013*). In this study, we showed that the identified *CRMP1* variants affect neurite outgrowth, which is a characteristic phenotype associated with many neurodevelopmental/psychiatric disorders (*Prem et al., 2020*). Our finding is in line with the reduced neurite outgrowth phenotype and impaired long-term potentiation in the *Crmp1⁻/⁻* mice (*Su et al., 2007*). Mechanistically, these cellular processes are regulated in a spatiotemporal manner through CRMP1 phosphorylation by several kinases such as Cdk5, Rho/ROCK, or GSK3, and any disturbances in these kinases or mutation of specific residues culminates in the abnormal brain development (*Yamashita et al., 2006*; *Su et al., 2007*; *Charrier et al., 2006*; *Makihara et al., 2016*; *Uchida et al., 2005*; *Yamashita and Goshima, 2012*). In the case of *Cdk5⁻/⁻* mice, abnormal dendritic spine morphology was shown in cortical neurons as it was in *Crmp1⁻/⁻* mice. Intriguingly, both mouse models show schizophrenia-like behavior, indicating the commonly shared cellular mechanism for dendritic development (*Su et al., 2007*; *Yamashita et al., 2013*) In the hippocampal slices of both *Crmp1⁺/⁻* and *Crmp1⁻/⁻ mice*, LTP induction was impaired (*Su et al., 2007*). This highlights the importance of normal expression of *Crmp1* for learning and memory in mice. In the context of neuronal development, defective phosphorylation of tyrosine residues in CRMP1 by Fyn-mediated Reelin signaling impairs cortical neuron migration (*Yamashita et al., 2006*). Over the last two decades, new phosphorylation-specific sites in CRMP1 are being reported and recently, phosphorylation of Tyr504 residue by Fyn has been shown to play an important step in Sema3A-regulated dendritic development of cortical neurons (*Kawashima et al., 2021*) Recently, phosphorylation of Crmp1 at Ser522 has been reported to play role in the ALS pathogenesis (*Asano et al., 2022*; *Kawamoto et al., 2022*). In this study, we showed that the reported *CRMP1* variants lead to lost-of-function and overexpression of wildtype *CRMP1* might be considered as a potential therapeutic strategy to rescue the phenotype.

In conclusion, we report for the first-time human *CRMP1* variants, link them to a human neurodevelopmental disease and highlight underlying pathomechanisms. Our report adds *CRMP1* to list of other *CRMP* genes linked to neurological disorders and underlines the important role of CRMP1 in the nervous system development and function. As a future perspective, it will be interesting to find other

*CRMP1* variants and elucidate the role of specific residues in various neuronal process and associate with the phenotypic spectrum of neurological diseases.

# Materials and methods

## Key resources table

| Reagent type (species) or resource | Designation | Source or reference | Identifiers | Additional information |
|---|---|---|---|---|
| Gene (*Homo sapiens*) | CRMP1B | Genbank | NM_001313 transcript variant 2, mRNA | Short isoform of human CRMP1 |
| Strain, strain background (*Escherichia coli*) | BL21 (DE3) pLysS | BioDynamics | Cat# DS260 Lot 0Q041 | |
| Genetic reagent (*M. musculus*, female) | ICR | Nihon SLC | RRID:IMSR_TAC:icr | |
| Cell line (*Homo sapiens*) | HEK293T | ATCC | CRL-3216 | Authenticated by STR analysis |
| Transfected construct (*Homo sapiens*) | pc3.1beta2-human CRMP1B-V5 wildtype | This paper | pc3.1b2-hCRMP1B-V5 | PCR-amplified from Invitrogen human brain cDNA library Available from F. Nakamura's lab |
| Transfected construct (*Homo sapiens*) | pc3.1beta2-human CRMP1B-V5 T313M | This Paper | pc3.1b2-T313M-V5 | Available from F. Nakamura's lab |
| Transfected construct (*Homo sapiens*) | pc3.1beta2-human CRMP1B-V5 P475L | This Paper | pc3.1b2-P475L-V5 | Available from F. Nakamura's lab |
| Transfected construct (*Homo sapiens*) | pGEX-human CRMP1B wildtype, T313M, P475L | This Paper | pGEX-hCRMP1Bwt, pGEX-T313M, pGEX-P475L | GST-fusion protein expression vectors Available from F. Nakamura's lab |
| Transfected construct (*Discosoma* sp.) | pENN.AAV.CAG.tdTomato.WPRE.SV40 | Gift from James M. Wilson (Addgene plasmid) | # 105554; RRID:Addgene_105554 | — |
| Antibody | Anti-MAP2 (Rabbit polyclonal) | Covance | PRB−547C RRID:AB_2565455 | 1:5000 |
| Antibody | Anti-Myc-tag (9E10, mouse monoclonal) conjugated agarose resin | BD | Cat# 631208 Lot 4100006 | 1:1000 |
| Antibody | Anti-Myc-tag (My3) (Mouse monoclonal) | MBL | Cat# M192-3 Lot 009 | 1:5000 |
| Antibody | Anti-V5 tag (Mouse monoclonal) | ThermoFisher | Cat# R960-25 RRID:AB_2556564 Lot 2311401 | 1:5000 |
| Antibody | Anti-V5 tag (Rabbit polyclonal) | Novus Biological | Cat# NB600-381 RRID:AB_527427 | 1:5000 |
| Antibody | Anti-mouse Immunogloblins/HRP (goat polyclonal) | GE | Cat# NA931V | 1:5000 |
| Antibody | Anti-mouse Immunogloblins/ biotin (goat polyclonal) | Jackson | Cat# P0260 | 1:5000 |
| Antibody | Anti-rabbit Immunogloblins/HRP (goat polyclonal) | GE | Cat# NA934V | 1:5000 |
| Antibody | Anti-rabbit Immunogloblins/biotin (goat polyclonal) | Vector | Cat# P0260 | 1:5000 |
| Recombinant DNA reagent | pc3.1beta2-V5 | *Kawashima et al., 2021* (J.Neurochem 157: 1207–1221 (2021)) | N/A | |
| Recombinant DNA reagent | pGEX-6P-1 | Cytiva | Cat# 28954648 | |

*Continued on next page*

*Continued*

| Reagent type (species) or resource | Designation | Source or reference | Identifiers | Additional information |
|---|---|---|---|---|
| Sequence-based reagent | Human CRMP1B-1f primer | This paper | PCR primers | atcgaattcgccATGTCGTACCAGGGCAAGAAGAGCAT |
| Sequence-based reagent | Human CRMP1-572r primer | This paper | PCR primers | atcctcgagACCGAGGCTGGTGATGTTGGAGCGGCCACCA |
| Sequence-based reagent | T313M mutation primer | This paper | Mutation primers | CCGGACCCTACCAtGCCCGACTACTTG |
| Sequence-based reagent | P475L mutation primer | This paper | Mutation primers | CGGAAGGCGTTCCtGGAGCACCTGTAC |
| Sequence-based reagent | CRMP1-forward primer (P1) | This paper | Sanger sequencing primers | TCTTCGAGGGTATGGAGTGC |
| Sequence-based reagent | CRMP1-reverse primer (P1) | This paper | Sanger sequencing primers | CGTCAGATCTCGATTCCCCA |
| Sequence-based reagent | CRMP1-forward primer (P2) | This paper | Sanger sequencing primers | ACAAAAGCGGATCCTGGAGA |
| Sequence-based reagent | CRMP1-reverse primer (P2) | This paper | Sanger sequencing primers | GTACACAGGGCAGTTGATCC |
| Peptide, recombinant protein | PreScission protease | Cytiva | Cat# 27084301 | |
| Peptide, recombinant protein | hCRMP1B-wt, T313M, P475L | This paper | | Purified from *E. coli* BL21 (DE3) pLysS cells Available from F. Nakamura's lab |
| Commercial assay or kit | QuickChange Multi Site-Directed Mutagenesis Kit | Agilent Tech | Cat #200515-5 | |
| Commercial assay or kit | Tyramide signal amplification (TSA) system | PerkinElmer | Cat No. NEL700A001KT | |
| Chemical compound, drug | Glutathione Sepharose 4B, 10 ml | Cytiva | Cat# 17075601 | |
| Chemical compound, drug | Poly-L-lysine | Wako | Cat No. 163-19091 | |
| Chemical compound, drug | Fugene-6 | Promega | Cat No. E2691 | |
| Software, algorithm | Fiji (2.0.0-rc-59/1.51n) | *Schindelin et al., 2012* Fiji software (2.0.0-rc-59/1.51n) | https://imagej.net/software/fiji/ | |
| Software, algorithm | Prism (Version 9.4.1) | GraphPad Software, LLC. | https://www.graphpad.com/scientific-software/prism/ | |

Written informed consent was obtained from all parents of the patients. The human study adhered to the World Health Association Declaration of Helsinki (2013) and was approved by the local ethics committees of the Charité (approval no. EA1/212/08). All animal experimental protocols were checked and approved by the Institutional Animal Care and Use Committee of the Tokyo Women's Medical University (protocol no. AE21-086).

## Genetic analyses

In family 1, DNA samples of family members were isolated from peripheral blood lymphocytes, and whole-genome sequencing (WGS) was performed on a HiSeq XTen Deep Sequencer (Illumina, CA, USA) according to the manufacturer's instructions. The QC values include: >Q20 sequences 95.4%, the average depth is 36×, the >20× coding exons coverage 96.5%. The primary data analysis was done by a combination of Burrows-Wheeler Alignment (BWA) sequence aligner for reads alignment to human reference genome GRCh37/hg19, Genome Analysis Toolkit (GATK) pipeline for calling single-nucleotide variants (SNV), and small insertions and deletions (indel), and ANNOVAR for variant

characterization (*Li and Durbin, 2010*; *Wang et al., 2010*; *DePristo et al., 2011*). Identified variants were filtered through comparison with the disease-associated variants in the known databases (Human Gene Mutation Database (HGMD, 2020.2) and the Online Mendelian Inheritance in Man (OMIM)) and polymorphism databases (dbSNP143, 1000 Genome). The population allele frequency cutoff: for de novo heterozygous mutations, we demanded that there should be no match in databases such as GnomAD or ExAc; for homozygous or compound heterozygous variants, we required that there should be no match of homozygote in databases such as GnomAD or ExAc, if there be heterozygous match in databases, the allele frequency should be lower than $1 \times 10^{-5}$. Sanger sequencing of the *CRMP1* gene (NM_001014809) was performed to confirm the identified variants in the patients and further family members (*CRMP1-forward primer: TCTTCGAGGGTATGGAGTGC, CRMP1-reverse primer: CGTCAGATCTCGATTCCCCA*) and performed segregation analysis of the variants with the phenotype.

In family 2, variants were detected by routine WES diagnostics and variant calling using a parent-offspring trio approach as described previously (*de Ligt et al., 2012*). Briefly, the exome was captured using the Agilent SureSelectXT Human All Exon v5 library prep kit (Agilent Technologies, Santa Clara, CA, USA). Exome libraries were sequenced on an Illumina HiSeq 4000 instrument (Illumina, San Diego, CA, USA) with 101 bp paired-end reads at a median coverage of 75× at the BGI Europe facilities (BGI, Copenhagen, Denmark). The datasets have been filtered with 5% population allele frequency. Sequence reads were aligned to the hg19 reference genome using BWA and variants were subsequently called by the GATK unified genotyper, version 3.2-2 and annotated using a custom built diagnostic annotation pipeline. Array analysis (CytoScan HD platform) detected no clinically significant copy number variation (CNV) and this was confirmed by follow-up routine WES-CNV analysis using the Conifer algorithm. Multispecies sequence alignment was performed using Multialin, PhyloP, and PhastCons and the variants disease-causative nature was predicted using the free online tool Mutation Taster (https://www.mutationtaster.org/).

In family 3, trio-based whole-genome sequencing was performed on blood-derived genomic DNA at the Laboratoire de biologie médicale multisites SeqOIA (LBMS SeqOIA, Paris, France). After extraction, DNA samples were sonicated, and libraries were prepared using the NEBNext Ultra-II kit (New England Biolabs). Whole-genome sequencing was performed in paired-end 2 × 150 bp mode, with a NovaSeq6000 (Illumina): Proband % genome >15× 96.3%; DOC: 37×, Mother % genome >15× 96.3% DOC: 37.2×, Father% genome >15× 96.4% DOC: 38.4×. Demultiplexed data were aligned to the genomic reference (hg19). Recalibration and variant calling were performed using GATK4 (Broad Institute). Short variants were annotated with SNPeff (4.3t). Copy Number Variants >4 kb were called using CNVnator (v0.4.1) and annotated with AnnotSV (v2.5.1). An average depth-of-coverage of >50× was obtained for both probands, and variants were prioritized according to impact, frequency, and segregation. The allele frequency cutoff applied was maximum 0.05. Sanger sequencing was performed to confirm the identified variants in the *CRMP1* gene (*CRMP1-forward primer: ACAAAAGC GGATCCTGGAGA, CRMP1-reverse primer: GTACACAGGGCAGTTGATCC*). Haploinsufficiency and dominant negative are likely to be applicable in the same gene, such as some of the genes encoding for aminoacyl-tRNA synthetases (*Meyer-Schuman and Antonellis, 2017*).

## Structural analysis of CRMP1

Structural interpretation of CRMP1 (PDBid: 4b3z) was performed and figures were generated using PyMOL (*Schrödinger LLC*). Sequence alignment was achieved using standard settings and structures from the PDB: 4b3z using Multalin and Espript (*Corpet, 1988*; *Robert and Gouet, 2014*).

## Construction of human CRMP1 expression vectors

The coding fragment of human CRMP1B, CRMP1 short isoform, was PCR-amplified with 5′-*atcg aattcgccATGTCGTACCAGGGCAAGAAGAGCAT*-3′ and 5′-*atcctcgagACCGAGGCTGGTGATGTTG GAGCGGCCACCA*-3′ primers from human brain cDNA library (Invitrogen) and cloned into pcDNA3.1 myc-His A expression vector, resulting pc3.1myc-CRMP1B-wildtype. The missense variants in the pedigree1 (g.4:5830253G>A, c.1766C>T, p.P589L (CRMP1A), p.P475L (CRMP1B)) or in the pedigree2 (g.4:5841279G>A, c.1280C>T, p.T427M (CRMP1A), p.T313M (CRMP1B)) were introduced to pc3.1myc-CRMP1B using QuikChange Multi Site-Directed Mutagenesis Kit (Agilent Tech, Cat No. #200515) with 5′-CGGAAGGCGTTCCtGGAGCACCTGTAC-3′ or 5′-*CCGGACCCTACCAtGCCCGA*

*CTACTTG*-3' primers, respectively. The resulting constructs were designated as pc3.1myc-CRMP1B-P475L and pc3.1myc-CRMP1B-T313M. These constructs were confirmed by cDNA-sequencing. The coding fragments of CRMP1B-wildtype, -T313M, and -P475L were transferred to pGEX6P1, a GST-tagged bacterial expression vector, and pc3.1beta2-V5, a neuronal expression vector harboring beta-actin promotor.

## Expression and purification of recombinant *hCRMP1* variants

Subcloned GST-tagged *CRMP1B*-V5 expression plasmids (wildtype, P475L, T313M) were transformed into *E. coli* BL21 (DE3) pLysS cells (BioDynamics Laboratory, Tokyo, Japan). These bacteria were harvested and added to 200-ml Terrific Broth medium (Invitrogen, Waltham, MA, USA) containing 100 µg/ml ampicillin (Sigma, St. Louis, MO, USA). After incubation for 1.5 hr at 37°C, the protein production was induced by 0.1 mM isopropyl β-D-1-thiogalactopyranoside (Sigma) for 18 hr at 20°C. The bacteria were collected by centrifugation and suspended in 24-ml homogenization buffer (H buffer; 50 mM Tris/HCl (pH 8.0), 100 mM KCl, 1 mM Ethylenediamine tetraacetic acid (EDTA), 5% glycerol, 1% NP-40, 1 mM dithiothreitol (DTT)) and then sonicated four times for 5 min on ice using Ultrasonic Homogenizer (Microtec Co, Ltd, Chiba, Japan). After centrifugation at 12,000 rpm for 30 min at 4°C, resultant supernatants were mixed with 400 µl of Glutathione Sepharose 4FF resin (Cytiva, Tokyo, Japan) and gently inverted for 24 hr at 4°C. The resins were collected and washed four times with 10 ml H buffer, and then washed five times with 1-ml cleavage buffer (C buffer; 20 mM Tris/HCl (pH 8.0), 150 mM NaCl, 0.1% NP-40, 1 mM DTT). The pelleted resins were suspended in 100 µl of C buffer including 10 U PreScission protease (Cytiva) and incubated for 24 hr at 4°C with constant agitation. These mixtures were transferred to the filter cups (Vivaclear Mini 0.8 µm PES; Sartorius, Stonehouse, UK), and the flow through fractions were obtained by centrifugation at 10,000 rpm for 1 min at 4°C. Protein concentrations were determined using the Bradford reagent (Thermo Fisher Scientific, Waltham, MA, USA), and the purified proteins were assessed by SDS–PAGE.

## Size-exclusion chromatography

The molecular weights of the recombinant hCRMP1s in the solution were estimated by size-exclusion chromatography using a HiPrep 16/60 Sephacryl S-300 HR column (Cytiva). The chromatography was operated using Acta prime plus (Cytiva) on the condition: Injection; 150 µg protein, flow rate; 0.5 ml/min, buffer; 50 mM sodium phosphate, 150 mM NaCl (pH 7.2) at 4°C. Fractions were collected at every 1 ml. Before sodium dodecyl sulfate–polyacrylamide gel electrophoresis (SDS–PAGE) analyses, 43–77th fractions were concentrated to 50 µl using Amicon Ultra Centrifugal Filter 10,000 MWCO (Merck Millipore, MA, USA). Prior to analyzing the samples, a standard protein mixture (Gel filtration calibration kit HMW; Cytiva) was chromatographed on the same condition and the molecular weights estimated from the chromatogram were indicated on the top of the gels.

## STR profile and mycoplasma contamination test for HEK293T cell line

For authentication of HEK293T cells used in the present study, 10 locus analysis of their DNA was outsourced (BEX Co, Ltd, Tokyo, Japan). In the database (https://jcrbcelldata.nibiohn.go.jp/str/database), there was no cell decided by identical (>0.8 EV), however, all listed cell lines showing high EV (>0.65) belonged to HEK293T cell lines, suggesting the gene mutations caused by the long-term passage, but not the contamination by other cell lines. Detection of contaminated mycoplasma in HEK293T was performed using MycoStrip (InvivoGen, San Diego, CA, USA) according to the manual. The conditioned medium obtained from 3.5-day culture did not show a positive band up to 50-min reaction, indicating no contamination of mycoplasma. Details of these results were displayed in the supplementary file.

## Immunoprecipitation

HEK293T cells were co-transfected with pc3.1beta2-myc-CRMP1B-wildtype and either one of V5-tagged CRMP1B constructs, wildtype, T313M, or P475L. After 2 days of incubation, cells were lysed with IP150 buffer (20 mM Tris–HCl, pH7.4, 150 mM NaCl, 5 mM EDTA, 1 mM NaF, 0.5 mM $Na_3VO_4$, 1% Triton X-100, protease inhibitors) and centrifuged at 13,500 rpm for 20 min at 4°C. The solubilized fractions were mixed with anti-c-Myc (9E10) agarose beads for overnight at 4°C. The beads were washed with IP150 buffer and analyzed on immunoblots with anti-V5 rabbit pAb (1:5000) and

anti-Myc mouse mAb (My3, 1:5000). horseradish peroxidase (HRP)-conjugated secondary antibodies were utilized for the detection of the primary antibodies on the membranes. The density of V5-CRMP1 band was analyzed with ImageJ (1.53K). The ratio of immunoprecipitated V5-CRMP1 and input in each condition was scored.

## Animals

Total number of pregnant ICR mice (RRID:IMSR_TAC:icr) utilized in this study was 6. All animal experimental protocols were checked and approved by the Institutional Animal Care and Use Committee of the Tokyo Women's medical University with protocol no. 'AE21-086'. All animal experiments were performed during daytime.

## Transfection to primary cultured mouse embryonic cortical neurons

Pregnant ICR female mice were purchased from Nihon SLC (Shizuoka, Japan). Overdose isoflurane (7–8%)/air mixture was inhaled to euthanize the pregnant ICR mice until the loss of breathing. The E15–16 mouse embryos dissected from the pregnant mice were immediately placed in ice-cold phosphate-buffered saline (PBS) and euthanized by decapitation. Cortices dissected from the embryos were treated with 1% trypsin/PBS for 5 min at 37°C and quenched with the addition of 0.5% trypsin inhibitor. The dissociated neurons were centrifuged at $800 \times g$ for 5 min at 4°C, washed once, and suspended in Dulbecco's Modified Eagle Medium (DMEM)–10% fetal bovine serum (FBS). The cells ($1.0 \times 10^6$) were transfected with 10 µg of pc3.1beta2-V5 harboring *CRMP1B*-wildtype, -P475L, or -T313M or tdTomato expression vector (pENN.AAV.CAG.tdTomato.WPRE.SV40, a gift from James M. Wilson, Addgene plasmid # 105554) using an electroporation equipment NEPA21 (NEPA GENE, Chiba, Japan). The condition of electroporation was follows: Poring pulse (275 V, 0.5 ms pulse length, 50 ms pulse interval, 10% decay, +pulse orientation, two times); Transfer pulse (20 V, 50 ms pulse length, 50 ms pulse interval, 40% decay, ±pulse orientation, five times). The cells were suspended in DMEM–10% FBS and seeded ($1 \times 10^4$/well) on a 24-well culture-plate coated with 0.05 mg/ml of poly-L-lysine (Wako, Cat No. 163-19091). After overnight incubation, the medium was replaced with Neurobasal medium supplemented with 2% B-27, 2 mM Glutamax, 50 U/ml penicillin, and 50 µg/ml streptomycin (500 µl/well) and incubated for 6–7 days at 37°C.

## Immunocytochemistry of mice cortical cultured neurons

The primary cultured cortical neurons were fixed with 4% paraformaldehyde/PBS for 30 min at 25°C, replaced with PBS, and stored at 4°C. As the expression of V5-*CRMP1B* constructs was limited, tyramide signal amplification (TSA) system (PerkinElmer, Cat No. NEL700A001KT) was applied for the detection of V5 signal. Briefly, the cells were treated with PBS containing 0.3% $H_2O_2$ for 20 min at 25°C and blocked with 1:1 mixture of PBS supplemented with 0.1% Triton X-100 (PBST) containing 10% skim milk and Tris-buffered saline supplemented with 0.1% Triton X-100 (TBST) containing 5% normal goat serum (NGS) for 30 min at 25°C. The cells were washed twice with TBST and incubated with the primary antibody mixture consisting of TBST 2.5% NGS, anti-V5 mouse mAb (1:5000) and anti-MAP2 rabbit pAb (1:5000, Covance (PRB-547C) RRID: AB_2565455) for overnight at 4°C. The cells were sequentially incubated each for 1 hr at 25°C with TBST 2.5% NGS containing biotin-conjugated anti-mouse secondary antibody (1:3000, Jackson) and TBST 2.5% NGS containing streptavidin–HRP conjugate (1/500, dilution, PerkinElmer). Then the cells were reacted with tyramide signal amplification mixture for 10 min at 25°C and subsequently incubated with TBST 2.5% NGS containing streptavidin–Alexa594 (1:2000) and anti-rabbit secondary antibody-Alexa488 for 2 hr at 25°C. The images of immunostained neurons were captured by an Olympus IX70 microscope equipped with ×10 Objective lens and DP74 camera. The neurite outgrowth of V5- or tdTomato-positive neurons was scored with Fiji software (2.0.0-rc-59/1.51n). In each condition, 13–20 V5- or tdTomato-positive neurons were analyzed.

## Statistical analysis

Data were graphed and analyzed using GraphPad Prism (9.4.1). The paired line graph (*Figure 2D*) is presented with individual values. The line connects the values for each condition from the same blot. Data were analyzed by one-way repeated measures analysis of variance (ANOVA) with assuming sphericity and followed by Tukey's post hoc test. Bar graph (*Figure 3E*) is presented as mean ± standard

error of the mean with individual values. Data were analyzed by one-way ANOVA followed by Tukey's post hoc test. Original data and statistical analysis are presented as supplemental data. Significance is presented as $*p < 0.05$, $**p < 0.01$, or $***p < 0.001$.

## Acknowledgements

The authors thank the index families for their participation in this study. We acknowledge Petra Bittigau, Akosua Sarpong for attending the patients; Anna Tietze for the analysis of cranial MR images; Sabrina Pommer and Jessica Fassbender for technical assistance; Piotr Neumann for assistance in data interpretation and lively discussions; Dr. Tanabe Kenji and Dr. Fumitoshi Saitoh for electroporation equipment. We acknowledge the funding resources for this study: German Research Foundation (SFB1315, FOR3004, AMK), Einstein Stiftung Fellowship through the Günter Endres Fond (AMK), the Sonnenfeld Stiftung (AMK, KLM), the Berlin Institute of Health (BIH, CRG1, AMK), the Charité (AMK, ER, LLB, KLM), and by grants-in-aid for Scientific Research from the Japan Society for the Promotion of Science (JSPS) (16K07062) (FN). We acknowledge financial support from the Open Access Publication Fund of Charité – Universitätsmedizin Berlin and the German Research Foundation (DFG).

## Additional information

### Funding

| Funder | Grant reference number | Author |
|---|---|---|
| Charité - Universitatsmedizin Berlin | | Lena-Luise Becker Ethiraj Ravindran Angela M Kaindl Konstantin L Makridis |
| Berlin Institute of Health | CRG1 | Angela M Kaindl |
| Japan Society for the Promotion of Science | 16K07062 | Fumio Nakamura |
| Sonnenfeld Stiftung | | Konstantin L Makridis |
| German Research Foundation | FOR3004 | Angela M Kaindl |
| German Research Foundation | SFB1315 | Angela M Kaindl |
| Einstein Stiftung Fellowship through the Günter Endres Fond | | Angela M Kaindl |
| National Natural Science Foundation of China | 81671067 | Hao Hu |
| Major Medical Collaboration and Innovation Program of Guangzhou Science Technology and Innovation Commission | 201604020020 | Hao Hu |
| National Natural Science Foundation of China | 81974163 | Hao Hu |

The funders had no role in study design, data collection, and interpretation, or the decision to submit the work for publication.

### Author contributions

Ethiraj Ravindran, Formal analysis, Writing – original draft, Project administration, Writing – review and editing, Visualization; Nobuto Arashiki, Kohtaro Takizawa, Formal analysis, Methodology; Lena-Luise Becker, Konstantin L Makridis, Maaike Vreeburg, Alexander PA Stegmann, Hao Hu, Formal analysis; Jonathan Lévy, Resources, Formal analysis; Thomas Rambaud, Na Li, Methodology; Yoshio Goshima,

Supervision; Bénédicte Demeer, Formal analysis, Writing – review and editing; Achim Dickmanns, Methodology, Writing – review and editing; Fumio Nakamura, Conceptualization, Formal analysis, Supervision, Funding acquisition, Writing – original draft, Writing – review and editing; Angela M Kaindl, Conceptualization, Formal analysis, Funding acquisition, Writing – original draft, Writing – review and editing

## Author ORCIDs
Ethiraj Ravindran ⓘ http://orcid.org/0000-0002-0095-116X
Konstantin L Makridis ⓘ http://orcid.org/0000-0002-2609-4557
Alexander PA Stegmann ⓘ http://orcid.org/0000-0002-9736-7137
Hao Hu ⓘ http://orcid.org/0000-0002-3567-609X
Fumio Nakamura ⓘ http://orcid.org/0000-0003-2029-6083
Angela M Kaindl ⓘ http://orcid.org/0000-0001-9454-206X

## Ethics
Written informed consent was obtained from all parents of the patients. The human study adhered to the World Health Association Declaration of Helsinki (2013) and was approved by the local ethics committees of the Charité (approval no. EA1/212/08).

All animal experimental protocols were checked and approved by the Institutional Animal Care and Use Committee of the Tokyo Women's medical University with protocol no. 'AE21-086'. All animal experiments were performed at daytime. The study was not pre-registered.

## Decision letter and Author response
Decision letter https://doi.org/10.7554/eLife.80793.sa1
Author response https://doi.org/10.7554/eLife.80793.sa2

---

# Additional files

### Supplementary files
• Supplementary file 1. Short tandem repeat (STR) profile and mycoplasma contamination test for HEK293T cell line.
• MDAR checklist

### Data availability
All data generated or analyzed during this study are included in the manuscript. Source Data files have been provided for Figures 2 and 3.

---

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
