## [Editor Report]

The authors report the first human disease-related variants in CRMP1 leading to a neurodevelopmental syndrome. After a successful first review, the authors have addressed the reviewer comments fully and have carefully assessed and fulfilled the criteria for essential revisions with the inclusion of new data. The results will be of importance to cell biologists and clinicians.

---

## [Decision Letter]

**Decision letter after peer review:**

Thank you for submitting your article "Monoallelic CRMP1 gene variants cause neurodevelopmental disorder" for consideration by *eLife*. Your article has been reviewed by 3 peer reviewers, one of whom is a member of our Board of Reviewing Editors, and the evaluation has been overseen by Jeannie Chin as the Senior Editor. The following individual involved in the review of your submission has agreed to reveal their identity: Ahmet Okay Caglayan (Reviewer #1).

The reviewers have discussed their reviews with one another, and the Reviewing Editor has drafted this to help you prepare a revised submission. Please note that the policy at *eLife* is that invitations for revised manuscripts typically include only requests for essential revisions for requested experiments that can be performed in a relatively short timeframe, i.e. within 6 weeks or so (see below). Other improvements suggested by the reviewers should be carefully considered, and attempts to address these will benefit the manuscript. Additionally, *eLife* provides full reviews including major weaknesses, which should be addressed in a point-by-point rebuttal. If you do not feel that you can address these issues within this timeframe, please let us know.

Essential Revisions:

1) The authors should include details on the population allele frequency cut-off they have used to filter the WES data. Regarding variant classification, appropriate ACMG rules should be applied to identified variants and implemented into the text.

2) Figure 2 Panel A does not show loading control. Panel B at 720 kDa band is not convincing. Results should be repeated with size exclusion chromatography and/or another method to determine molecular weight and should be quantified from triplicate experiments. Panel C is also not convincing and should be repeated to more carefully show results, and quantified.

*Reviewer #1 (Recommendations for the authors):*

This is an interesting manuscript examining mutations in CRMP1 and their linkage to a rare syndrome in humans first time in the literature. Overall, the authors achieved their aim of demonstrating the existence of a novel neurodevelopmental disease-gene association and implicating CRMP1 as an important player in human neurodevelopment. This is of significant importance to the medical field, human geneticists, and neuroscientists. Further clarification of the disease mechanism is needed for therapeutic considerations, i.e. whether gene augmentation therapies might rescue the phenotype (loss-of-function/haploinsufficiency) or if allele-specific knockdown or gene editing is necessary (dominant negative).

However, there are some other issues to be addressed:

1) Please include primers for Sanger sequencing for P1 and P3. Adding representative chromatograms of Sanger sequencing is encouraged.

2) Since a gain-of-function mechanism is provided and often gain-of-function mutants are generated by expressing a gene at a much higher level than normal in cells, it is recommended to analyze gene expression abundance of CRMP1 in heterozygous variant carriers and non-carriers if tissue/cell samples are available.

3) Pathogenic mechanism of CRMP1 variants for all symptoms is not well documented.

4) Laboratory data are not sufficient. The second and third patients show obesity. Metabolic and endocrinological surveys should be included if possible.

5) Details on CNV analysis (SNP-array, WES-based CNV analysis, other? or not performed) should be given for patient 2.

6) Since WES was performed, more details on the QC values of WES (percent of exome captured, percent of exome analyzed, depth of coverage, etc.) should be included. This information is especially important for patient 1 where the WES coverage is lower than usual. There is one line about this as part of the study design of the manuscript for patient 3, but that does not provide sufficient information.

7) The authors should include details on the population allele frequency cut-off they have used to filter the WES data.

8) Regarding variant classification, appropriate ACMG rules should be applied to identified variants and implemented into the text.

9) Interpreted variants may be deposited in ClinVar.

10) P475L and T313M mutations selected for functional studies: why were short isoforms of the CRMP1 gene chosen instead of canonical ones?

11) For example, on Page 8 Line 142 "…CRMP1B-V5" and on Page 15 Line 311 "…P475L and T313M mutations affect homo-oligomerization of CRMP1B" How is CRMP1B different than CRMP1? This should be explained for the readers' sake.

12) Page 16 Line 318 "…cleaved form, respectively". What is the shorter 60kd "truncated" form of the protein? Is the truncated form naturally occurring, some self-truncation due to bacteria expression and processing via protease? When expressed in HEK293, the lower, monomeric band is unique, there are not two different bands.

13) The manuscript needs to be carefully edited and corrected before publication to improve the clarity and quality of the paper. Editing by an author who is fluent in English might be of great benefit.

*Reviewer #2 (Recommendations for the authors):*

Another experiment that might strengthen the dominant-negative interpretation is co-localization studies, e.g. if variant CRMP1 disrupts the localization of wild-type CRMP1. Has this been done?

Do heterozygous knockout mice have any phenotype?

There should be more details about the cases identified. Genematcher? Other matchmaking platforms? Personal communications?

While the numbers are likely insufficient to show a statistically significant excess of de novo variants in CRMP1, ideally there would be details regarding the number of trio exomes/genomes performed to date at each site and the total number of de novo CRMP1 variants.

---

## [Author Response]

Essential Revisions:1) The authors should include details on the population allele frequency cut-off they have used to filter the WES data. Regarding variant classification, appropriate ACMG rules should be applied to identified variants and implemented into the text.

We thank reviewers for the comment. As per reviewer’s suggestion, we have included the details on population allele frequency cut-off applied to filter the WES data in the ‘Materials and Methods’ section in the revised manuscript.

2) Figure 2 Panel A does not show loading control. Panel B at 720 kDa band is not convincing. Results should be repeated with size exclusion chromatography and/or another method to determine molecular weight and should be quantified from triplicate experiments. Panel C is also not convincing and should be repeated to more carefully show results, and quantified.

We thank reviewers for the crucial comment on our experimental data of Figure 2. As per reviewer’s suggestions, the loading control of Figure 2A is shown in the raw data. We performed size exclusion chromatography and presented the results in the revised manuscript and discussed accordingly in page 23-24.

Reviewer #1 (Recommendations for the authors):This is an interesting manuscript examining mutations in CRMP1 and their linkage to a rare syndrome in humans first time in the literature. Overall, the authors achieved their aim of demonstrating the existence of a novel neurodevelopmental disease-gene association and implicating CRMP1 as an important player in human neurodevelopment. This is of significant importance to the medical field, human geneticists, and neuroscientists. Further clarification of the disease mechanism is needed for therapeutic considerations, i.e. whether gene augmentation therapies might rescue the phenotype (loss-of-function/haploinsufficiency) or if allele-specific knockdown or gene editing is necessary (dominant negative).

We thank reviewer for addressing on clarification for disease mechanism. We have performed experiments and included in the revised manuscript on page 23-25 and discussed in the ‘discussion’ section on page 27.

However, there are some other issues to be addressed:1) Please include primers for Sanger sequencing for P1 and P3. Adding representative chromatograms of Sanger sequencing is encouraged.

The electropherogram traces for P1 and P3 are presented as figure supplement 1 and the primer sequences are provided in the ‘Subjects and Methods’ section.

2) Since a gain-of-function mechanism is provided and often gain-of-function mutants are generated by expressing a gene at a much higher level than normal in cells, it is recommended to analyze gene expression abundance of CRMP1 in heterozygous variant carriers and non-carriers if tissue/cell samples are available.

It is an interesting comment from the reviewer, but unfortunately, we could not perform the experiment as we did not have access or permission to get samples from patients.

3) Pathogenic mechanism of CRMP1 variants for all symptoms is not well documented.

We revised the manuscript with additional experiments and show that may be the loss-of-function of the CRMP1 variant contributes to the symptoms reported in the index patients.

4) Laboratory data are not sufficient. The second and third patients show obesity. Metabolic and endocrinological surveys should be included if possible.

A routine metabolic screening for patients (P2) showed normal results and BMI was reported in the revised manuscript on page 20, line 376. We have included the metabolic and endocrinological findings of P3 in the revised manuscript on page 21.

5) Details on CNV analysis (SNP-array, WES-based CNV analysis, other? or not performed) should be given for patient 2.

As per reviewer’s suggestion, we have included the details in the revised manuscript on page 12.

6) Since WES was performed, more details on the QC values of WES (percent of exome captured, percent of exome analyzed, depth of coverage, etc.) should be included. This information is especially important for patient 1 where the WES coverage is lower than usual. There is one line about this as part of the study design of the manuscript for patient 3, but that does not provide sufficient information.

As per reviewer’s comment, we included the details of genetic analysis for patient 1 and 3 under ‘Subjects and Methods’ in the revised manuscript.

7) The authors should include details on the population allele frequency cut-off they have used to filter the WES data.

We included the details of population allele frequency cut-off we have used to filter the WES data in the revised manuscript.

8) Regarding variant classification, appropriate ACMG rules should be applied to identified variants and implemented into the text.

We included the details of population allele frequency cut-off we have used to filter the WES data in the revised manuscript.

9) Interpreted variants may be deposited in ClinVar.

We have deposited the variants in ClinVar.

10) P475L and T313M mutations selected for functional studies: why were short isoforms of the CRMP1 gene chosen instead of canonical ones?11) For example, on Page 8 Line 142 "…CRMP1B-V5" and on Page 15 Line 311 "…P475L and T313M mutations affect homo-oligomerization of CRMP1B" How is CRMP1B different than CRMP1? This should be explained for the readers' sake.

We thank reviewers for suggesting to bring clarity to the manuscript. We included the text in the revised manuscript on page 23. CRMP1 (collapsin response mediator protein 1) was originally identified as 572AA protein. Long form CRMP1 (686AA), a splicing variant at N-terminal region, was subsequently identified. These splicing variants have been designated as CRMP1B and CRMP1A, respectively. CRMP1B is the major isoform expressed in the nervous system. Most of CRMP1 studies including X-ray structural analysis have been performed with CRMP1B, short isoform of CRMP1. Therefore, we examined this analysis with CRMP1B isoform. In addition, we modified the figure 1B in the manuscript.

12) Page 16 Line 318 "…cleaved form, respectively". What is the shorter 60kd "truncated" form of the protein? Is the truncated form naturally occurring, some self-truncation due to bacteria expression and processing via protease? When expressed in HEK293, the lower, monomeric band is unique, there are not two different bands.

The truncated shorter 60kDa is probably self-truncation due to bacterial expression and processing via protease. We do not observe such truncation of transiently expressed CRMP1 in HEK293T cells.

13) The manuscript needs to be carefully edited and corrected before publication to improve the clarity and quality of the paper. Editing by an author who is fluent in English might be of great benefit.

Reviewer #2 (Recommendations for the authors):Another experiment that might strengthen the dominant-negative interpretation is co-localization studies, e.g. if variant CRMP1 disrupts the localization of wild-type CRMP1. Has this been done?

One possible experiment is to evaluate the effect of CRMP1B variants on endogenous CRMP1A localization in primary cultured neurons. However, no specific CRMP1A antibody is commercially available yet, so no experiments have been carried out.

Do heterozygous knockout mice have any phenotype?

In the Crmp1^+/-^ mice, LTP induction was impaired as observed in the Crmp^-/-^ mice. We have included the details about heterozygous mice under ‘Discussion’ in the revised manuscript.

There should be more details about the cases identified. Genematcher? Other matchmaking platforms? Personal communications?

We identified patient 2 and 3 via GeneMatcher. We are in contact with one more GeneMatcher profile, but unfortunately the consent of patient’s involvement in the study is not yet confirmed and could not consider for this study.

While the numbers are likely insufficient to show a statistically significant excess of de novo variants in CRMP1, ideally there would be details regarding the number of trio exomes/genomes performed to date at each site and the total number of de novo CRMP1 variants.

We have, since the beginning of routine WES diagnostics in 2011, not reported any other de novo CRMP1 variant. We performed a re-check in a cohort of over 20,000 exomes in which NO de novo CRMP1 variants are present, indicating that it is very rare. In addition, we searched in the denovo-db (https://denovo-db.gs.washington.edu/denovo-db/) and the Gene4Denovo database (http://www.genemed.tech/gene4denovo/search?inputValue=CRMP1&searchType=geneSymbol). We found 6 DNM in the denovo-db but all were in intronic regions. We found 31 DNM in the Gene4Denovo database, among which there were 5 missense mutations identified in patients of ASD or DD.